# Containerized Architecture Performance Analysis for IoT Framework Based on Enhanced Fire Prevention Case Study: Rwanda

**DOI:** 10.3390/s22176462

**Published:** 2022-08-27

**Authors:** Eric Hitimana, Gaurav Bajpai, Richard Musabe, Louis Sibomana, Kayavizhi Jayavel

**Affiliations:** 1African Centre of Excellence in the Internet of Things, University of Rwanda, Kigali P.O. Box 3900, Rwanda; 2Department of Computer and Software Engineering, University of Rwanda, Kigali P.O. Box 3900, Rwanda; 3National Council for Science and Technology, Kigali P.O. Box 2285, Rwanda; 4Department of Networking and Communications, School of Computing, SRM Institute of Science and Technology, Kattankulathur 603203, Tamil Nadu, India

**Keywords:** containers, DevOps, performance analysis, IoT, Raspberry Pi, fire prevention

## Abstract

Nowadays, building infrastructures are pushed to become smarter in response to desires for the environmental comforts of living. Enhanced safety upgrades have begun taking advantage of new, evolving technologies. Normally, buildings are configured to respond to the safety concerns of the occupants. However, advanced Internet of Things (IoT) techniques, in combination with edge computing with lightweight virtualization technology, is being used to improve users’ comfort in their homes. It improves resource management and service isolation without affecting the deployment of heterogeneous hardware. In this research, a containerized architectural framework for support of multiple concurrent deployed IoT applications for smart buildings was proposed. The prototype developed used sensor networks as well as containerized microservices, centrally featuring the DevOps paradigm. The research proposed an occupant counting algorithm used to check occupants in and out. The proposed framework was tested in different academic buildings for data acquisition over three months. Different deployment architectures were tested to ensure the best cases based on efficiency and resource utilization. The acquired data was used for prediction purposes to aid occupant prediction for safety measures as considered by policymakers.

## 1. Introduction

Software development challenges have arisen due to increases in IoT device-as-a-service, IoT infrastructure-as-a-service, and edge infrastructure-as-a-service [1]. In IoT, smart sensing and actuating objects are coupled with information and communication technologies (ICT). They are embedded in digital environments to produce huge volumes of data. These data need to be extracted, processed, and analyzed efficiently to arrive at meaningful information to support IoT applications.

The implementations are done using different platforms as services. The applicability levels of these platforms are challenged on their efficiency, performance, and variating scalabilities. This occurs in case of a system escalation, along with data analytics and resource consumption—commonly characteristics of monolithic applications [2]. A monolithic application is defined as a single-tiered software application that integrates users’ interfaces and data access codes as a single program within the same platform.

Nowadays, IoT devices contain embedded networking stacks, connecting directly to both publisher and subscriber service providers. The health and fire prevention domains are particularly related to the IoT reference architectures, focusing on sensors, network aspects, applications, and presentation aspects, including data analytics. IoT analytics are evolving with newer technologies, tools, and data processing, along with analytic methods. This has led to enhancements in the development of environments, with increased productivity and improved efficiency.

However, there is no ownership in data management, and services escalate out of control if service charges are not clear. The other issue is the implementation of an IoT information system, which requires a high-level holistic approach. That approach must mix traditional data collection from vendor-specific cloud back ends with embedded hardware and mobile devices in real-time.

IoT systems provide important, sometimes critical services, requiring uninterrupted operation or high levels of availability. Additionally, their complexity makes them difficult to integrate, while also complicating in-depth monitoring to prevent anomalies and failures [3].

The DevOps methodology maintains system characteristics, such as availability, scalability, and fault tolerance, on a live system [4,5,6,7,8]. The continuously rapid integration of new devices with feedback acquisition to instrument maintenance and updates leads to job accumulation. This methodology requires a multidisciplinary collaboration between development and operations teams to facilitate customer satisfaction. The collaborative team must test the platform in production, and build new releases to deploy them [9]. DevOps is a solution to facilitate that collaboration by accelerating and improving the cycles of maintenance [10,11,12].

The motivation of this research was to contribute to the improvement of building safety prevention (specifically, from fire outbreaks) using IoT technologies. The main question was how to integrate various IoT sensor nodes with scalable characteristics to provide a sound IoT system that provided real-time communication to the stakeholders (fire brigades) in case of a fire outbreak. The idea arose due to various challenges, such as IoT applications’ scalability, updatability, integration, and maintainability that affect performance and efficiency.

This research was aimed at designing and implementing an IoT framework architecture that supported multiple concurrent deployed IoT applications in a smart building. The main objective was to develop a containerized architecture performance analysis for the IoT framework based on enhanced fire prevention, tested in Rwanda. The contributions of this research were (1) the investigation of a detailed, state-of-the-art integration of recent techniques in the IoT implementation domain; (2) the proposal of layered IoT architecture to provide central and efficient management of connected sensor nodes, data, and intervening applications, while meeting final requirements needed; (3) the application of different software engineering paradigms, including microservices and container-based virtualizations in the implementation; (4) validation of the proposed framework in a smart building scenario in which the main concern was to improve security; (5) a performance evaluation, carried out to guarantee a scalable platform.

The rest of this paper was organized in the following order: Section 2 details related works, and Section 3 describes the materials and methods. Section 4 describes the proposed architecture. Section 5 highlights the design of the case study. Section 6 shows the implementation results with discussions. Section 7 discusses the performance evaluation of the proposed solution through the selected case study. Section 8 concludes the paper and details future works.

## 2. Related Works

The three-layer classic IoT reference architecture was extended by applying context-aware, serverless, microservices-based [13] cloud-centric techniques for IoT and people applications. Here, the focus was on frameworks that offered a high separation of concern degrees by splitting the application layer into different sub layers based on their responsibilities. It also traced atomic components to serverless microservices in order to facilitate the design, development, and deployment of applications in the healthcare domain.

### 2.1. Usage of DevOps Methodology

The three-tiered DevOps model was used to build an information system capable of providing real-time analytics [14]. Management within a smart city project was proposed to support the heterogeneous environment. However, the data visualization was conducted on an existing open-source platform, namely KIBANA from Elastic-search. The integration of DevOps into the design of IoT frameworks [5] presented an ongoing effort toward provision, with validation of solution security and privacy, along with the quality of complex software systems. The solutions were built on top of an established ENACT IoT DevOps concept and framework.

The development of a service model structure consisted of several functions within its services. The defined hardware compatibilities [15] tested an automation system in a building for emergency evacuation services. However, the concerns of continuous integration using DevOps techniques were not considered. The development of service contracts for IoT microservices from DevOps perspectives was considered to address the diversity of service contracts using common languages for IoT data and programming [16].

### 2.2. Application of Microservices

Rapidly developing technologies enable ubiquitous connectivity. This has resulted in the continuous growth of connected devices. On the other hand, it has also revealed challenges related to the development and architecture of information systems. The wide variety of proposed solutions enabled by interacting with devices makes it practically impossible to accurately manage the predicted ratio of data traffic patterns.

In the management of a growing amount of delivered services [17], it is essential to implement scalability mechanisms. Specific strategies were proposed herein to maintain system stability and scalability by ensuring the basic scaling mechanism (such as vertical and horizontal). Vertical scaling requires the addition of resources to a single processing node, resulting in more work handling with additional capacities. It is, however, limited by cost-effectiveness, physical constraints, and the availability of specialized hardware. Horizontal scaling requires a set of design practices and planned activities, leading to an inherent distribution within the system [18]. The horizontal scaling approach defines the unique ability as the keystone for large-scale architecture design.

The initial concepts for software developments used monolithic architecture [19]. The services were developed on a unique repository shared among multiple developers. In these concepts, developer teams had to guarantee that all other services would continue working as they did before. Another disadvantage was the presence of new updates, ready for deployment in the production environment. Updates require that all services be restarted, causing severe issues, from the users’ perspective. One final challenge is that each time one part/function fails, the entire set of services gets compromised.

Due to the wide variety of scenarios created in IoT and their different challenges (such as the heterogeneity of IoT devices, communication, and platforms supporting large numbers of connected devices), the monolithic pattern is not recommended [20].

The microservices-based architecture proposed the distribution of the applications in a set of services in such a way that each was independent of the others. These new patterns have been used to solve specified issues with new features, such as scalability and reusability [21]. The use of an IoT agnostic architecture, highlighting the role of the IoT platform used a system with a broader ecosystem of interconnected tools, aiming at increasing scalability, stability, interoperability, and reusability. Herein, the solution used microservices architecture and serverless computing applied to smart solutions.

Microservices, at a high level, can be considered a black box that has been divided into layers. Depending on the tasks to be performed, the layers can be reorganized. Microservices communication is possible through interfaces exposed by the Microservices themselves. Hence, each microservices offers an application programming interface (API) to facilitate different types of connections.

Microservices are useful for the development of IoT-based platforms. They allow services-oriented development, which is growing in popularity due to growing interest in parallel computation. The microservices architecture, designed and built using the Jolie programming language, was developed to work directly with a service-oriented paradigm [22]. Here, the prototype platform for supporting multiple concurrent applications for smart buildings was tested using an advanced sensor network, as was a distributed microservices architecture [21].

### 2.3. Container Management

Most of the capabilities of containers are first executed on constrained devices with limited and relatively inexpensive storage and computing resources, like Raspberry Pi [23], in order to provide the capability of transversal interoperability among different networks, integrating devices with limited capacity. Container-based services provisioning has demonstrated diverse benefits that allow applications to run on a diverse set of devices, with heterogeneity in terms of hardware, software, and network. To improve the performance of the containers in the described constrained devices within an environment, better solutions are needed to simplify and improve their management.

At this layer of the architecture, it is necessary to ascertain the performance of container schedulers or brokers, along with allocation among resources. Multiple containers, running in a cluster, support management activities. A container scheduler in a cluster has multiple goals, such as ensuring the use of resources efficiently; user access is restricted based on work and location. Additionally, it ensures that applications are quickly scheduled to avoid massive waiting times. It provides a high degree of balance between resources. It also provides an error-handling mechanism that ensures the most convenient allocation, both time-wise and power-wise [24]. The existence of diverse container schedulers for clusters supports the notion that there is no single solution to all problem paradigms. Among these schedulers, the following can be highlighted: Apollo Microsoft, Aurora Twitter, Borg Google, Fuxi Alibaba, Kubernetes Google, Omega Google, Swarm Docker, and YARN Apache, to name a few [25].

There are two key techniques used by containers to provide isolation of each container during execution environments. Those techniques are namespace and cGroup [26]. Namespace is used for isolation between execution environments, including process trees, networks, user IDs, and file systems of containers. cGroup allows for the management of resources, such as CPU, memory, network, and file I/O for containers. In the case of cGroup, resources can be controlled by assigning specific bounds to containers. It has also been shown that a specific number of resources for a container can be allocated to prevent resource contention between containers running concurrently.

The implementation and administration of diverse container-based applications are rapidly developing nowadays. This development could facilitate physical or virtual host distribution, built on top of central components, such as containers’ schedulers in general, and Dockers containers’ schedulers.

### 2.4. Docker Swarm

Among container schedulers, this research used Docker Swarm to address the problem of container management. Docker Swarm is a container management solution under the development of Docker Organization. It provides a standard Docker API, and its framework consists of two main components; first, a node that acts in the role of administrator executes an image of Swarm responsible for allocation of the containers on the remote machines (called agents, or nodes); secondly, the nodes with a remote Docker API capability are available to the administrator after the verification of its port correctness if it is open by the time the Docker daemon [27] starts.

Docker Swarm has different major scheduling strategies that ensure the possibility of the scheduler selecting a node for executing a container. It includes strategy name or direct node selection, and selection of the node with the lower set of containers in execution spread. Regardless of the load of each container, or binpack, the selection of the most packaged node (lower available CPU/RAM) and the random selection of the node are also included. In the latter case, if the previous strategies select several nodes according to the previous criteria, the scheduler selects a target node among them randomly.

The resource allocation strategy must be determined at the time of setup by the administrator or the spread strategy will be used by default. However, the spread method is the most extended scheduling strategy in most applications on the Docker Swarm Mode Cluster.

In this research, apart from microservices concepts which were used together, the architecture had additional containerization mechanisms built on top to ease scalability, flexibility, and transparency. This mechanism was achieved using visualization in a container-based approach, through Docker. The proposed concept allowed for the robust development of a framework operating in a distributed way, with greater speed and increased independence over underlying operating systems.

## 3. Materials and Methods

This section details the materials and methods used in the implementation of the proposed architecture in Section 4. The materials were classified into hardware and software components, data formats, and performance metrics.

### 3.1. Underlying Hardware Components

This subsection details different hardware components used during the implementation of the proposed framework architecture. The choice of criteria for each component depended on its availability and openness.

#### 3.1.1. Embedded Board

ARM embedded architecture uses lower-power characteristics [28]. The single Board Computer (SBC) family Raspberry Pi (RPi) takes advantage of these characteristics. RPis are small SBCs with 4 Advanced RISC Machine (ARM) Cortex-A53 1.2 GHz CPU and 1 GB RAM, with Wi-Fi integrated [29].

In this research paper, easily obtainable off-the-shelf hardware was used to facilitate reproducible results. The setup was composed of three RPi Alarm ARMv7 for Broadcom [30].

Table 1 describes the configurations of the experimental testbed. The desktop simulated the fog node while the RPis acted as the edge node and gateway.

#### 3.1.2. Edge Sensor Nodes

In fire detection, prevention, and occupant detection use cases, this paper used a couple of sensors to gather specific parameters. In a specific room, sensors were used to detect the presence of fire temperature, humidity, CO_2_, and smoke. Motion sensors were used in conjunction with obstacle sensors to detect the presence of human beings. Table 2 details the sensors used and their operating ranges based on the limits of the value of sensitivity and distance they cover.

#### 3.1.3. Network Connectivity and Protocols

The main tasks of a node were split into sensing/actuation, computation, storage, and communication, among the other nodes [31]. The node architecture was a bundle of multiple sensors which gathered environmental parameters to be sent to the cloud. The controller was the part of the node that fetched data from the sensor, exchanged them with the communication device, and controlled the behavior of the actuators. The microcontroller added the ability to interact with it by programming it. It had memory, enabling it to control the behavior of the node [32].

The implementation used the RPi board, which allowed LAN internet connectivity. The communication used two protocols: MQTT and HTTP. MQTT operated through data messaging architecture that used the publish/subscribe paradigm to ensure that the data was sent smoothly if the sensor node was subscribed to the broker and the database. At the end-user data access, the web services used HTTP so the user could access the information through a browser application.

### 3.2. Underlying Software Components

This section details the software components used in the implementation of the proposed framework, including the Docker engine as services containers manager, server setup for web and database management, data formats, protocols for transmission, services patterns, and performance monitoring tools.

#### 3.2.1. Dockers

Containerization is a technique used to abstract applications from the environments that they execute. Docker containers wrap up the software and its dependencies into a standardized unit for software development that includes everything it needs to run [33]. Containers are a lightweight alternative to virtual machines (VMs) for multi-tenancy within a single host. Here, computing devices were modeled as containers and managed using the Docker automation framework. This was accomplished in two parts: firstly, for resource allocation, and secondly, for software configuration.

The key advantages of using Docker for distributed applications included lower CPU overhead, version control, portability, and network performance improvement [34]. Docker developed a distributed service platform by considering the fault tolerance of services [35,36]. Dockers-based gateways and architecture have flexibility with scalability through the microservices paradigm [22,37].

This research took advantage of Docker’s features. The idea was to combine Docker container services and microservices strategies for modular, scalable edge computing frameworks for IoT smart buildings.

#### 3.2.2. Data Management and Data Format

All sensor-generated data were periodically saved in the database, with data management conducted on the webserver. The database module was implemented using PostgreSQL over the webserver used as Nginx.

Postgres was chosen as an object-relational database management system (RDBMS). It emphasized extensibility and standards compliance, and had the ability to provide replication of the database for scalability and security.

Nginx is an open-source reversed proxy server for HTTPS, with a load balancer. Nginx was chosen as the web server for its security, reliability, and load balancing when there was huge traffic with connected users. Its characteristics focused on high concurrency, high performance, and low memory usage.

The data exchange from sensor nodes to the cloud was formatted to allow efficient data transmission with low bandwidth. The data were converted into JSON (JavaScript Object Notation). JSON has been adopted as a standard text-based format for representing structured data.

#### 3.2.3. Application Services: Microservices

The proposed architecture in this research targeted the microservices architecture applied to the IoT framework. The architecture is made by different components including all real-time data acquisition and processing architectures in the IoT domain as shown in Figure 1. It had three key components: (1) data sources, (2) data processing, and (3) end-user data access.

The proposed solutions of microservices for smart buildings in general and fire detection with occupancy detection are detailed in Figure 2. 

The concept considered allowed independent mounting and unmounting of the new nodes. Every node was considered one service for data acquisition and transmission. It allowed any connected node to be updated and integrated with the management of the services manager played by Docker Swarm. The transmission protocol, data management, web services, and visualization services were managed as microservices. This allowed services reusability without affecting the whole structure.

#### 3.2.4. Monitoring Parameters

The performance of edge computing applications followed different runtime variations depending on the running conditions. Conditions could have included the number of arrival requests and the availability of virtual resources. Additionally, the network connection quality between different interoperating application components distributed over the communication network was an additional key to consider.

A mechanism to allow the aforementioned resources to be intelligently provisioned with little or no effort an application provider was required. The container manager was configured to ensure that the orchestration process of provisioning resources was done. The resources considered for provisioning in edge/cloud computing were: the usage of CPU, memory, storage, and network.

#### 3.2.5. Performance Analysis and Monitoring Tools

There are four levels of monitoring (containers, virtual machines (VM), end-to-end link quality, and application) for self-adaptative applications to edge computing applications [38]. Due to the heterogeneity of the applications used in this implementation context, the need for performance measurement was critical. This research was implemented for container-based performance monitoring purposes. It assisted the process of pulling and migrating container images across computing nodes in fog or cloud in contrast to virtual machine [39].

Docker itself had a built-in mechanism to access and convert container metrics into statistical format using Docker stats methods. Those metrics included runtime metrics and resource usage for a given container. This research proposed to use external built-in components to retrieve detailed sets of metrics by accessing exposed remote API using GET methods.

In monitoring the system behaviors, the following tools were used:-**Container advisor**, also known as cAdvisor, is a provided open-source platform. It was used to track, measure, aggregate, process, and display performance monitoring data for the running containers.-**Prometheus** is a monitoring tool relying on an open-source scheme. On top of monitoring activity, Prometheus can also store persistence time-series-based data for further processing and retrieval using PromQL language.-**Grafana** is a web-based open-source user interface, used to visualize large-scale performance monitoring information. It can be configured to work with the Prometheus database to visualize the time series data of the metrics.

## 4. Proposed Architecture

The proposed framework architecture consisted of a set of services distributed across different computing sites. Each application was made of separate independent deployable components/containers, with all possible functionalities hiding their implementation details. The provision of those separate modular services enabled the services to be characteristically small and loosely coupled. This improved both testability and management, using containerization packages configured in each end device node. Those configurations provided minimum environmental capabilities for the application to run properly.

Different layers of the proposed architecture are shown in Figure 1, describing each layer with its computational resources. The device layer known as the edge layer held all capabilities of sensing and communicating with the surrounding environment. The sensor nodes were grouped to allow connection and disconnection management.

The gateways worked cooperatively to ensure that the sent data were well cached and processed to reduce system response times. Additionally, the middle layer orchestrated communication between sensing nodes by providing a single-entry point to the local systems. The manager containers worked hand-in-hand with the device groups to maintain system updates of the resources. The manager containers were hosted in the cloud layer to allow multiple local domains to share data and resource status and make all resource-intensive applications run concurrently.

This proposed scheme was designed such that, in highly dynamic distributed systems, the management of multiple deployed sensor nodes and their resources could be achieved efficiently. The role of the manager (Docker Swarm), shown in Figure 2, was to ensure the most efficient resource allocation possible by keeping track of service deployment. It was able to manage sensor node groups and allow system scale-up, and updated as many containers as possible in clusters of machines.

The communication of sensitive data generated by multiple sources occurred in high velocity at high volumes. It was managed by messaging hub techniques provided by the MQTT (message queuing telemetry transport) protocol. The aforementioned multiple layering entry allowed a manager to define communication channels for applications. It reduced the intra-service network overhead. The message broker module was included at each layer in the Docker container.

### 4.1. Edge Layer

The sensing layer consisted of different sensor nodes deployed in the testing environment. The sensor nodes included, but were not limited, to: Temperature, Humidity, CO_2_, Proximity Sensor, and IR sensor. They were all deployed to a powered RPi board connected to a LAN network. Each RPi board was configured with a Docker machine and all sensor services were deployed on it in a container-based approach.

### 4.2. Gateway Layer

This layer was designed to manage the device groups deployed in different buildings to ensure the security and scalability of the framework. Once the new device was mounted, its corresponding gateway could update the container manager in the next layer (Fog layer) to update its discovery service, thus allowing other containers to be updated.

The gateway layer ensured that all containers were running efficiently. The decisions with light computation needs were also done at this level, as this Pi board had constrained resources.

### 4.3. Fog Layer

In a large-scale container deployment over numerous servers, tracking all the containers and managing their life cycles is a complex task. The fog layer was composed of a configured server with a machine-learning Linux operating system (OS). On top of the OS, the Docker Swarm was installed. Its role was to ensure the proper administration of service lifecycles in massive deployments of Docker containers. The layer allowed data management as well. In order to maintain the system, the security-with-redundancy paradigm, database server, and access were all done at this layer. All deployed services in the Docker environment cooperated hand-in-hand with the Docker hub to update new versions of container images used, just in case.

### 4.4. Cloud Layer

The research took advantage of cloud services by allowing all the public management and access to the implemented microservices resources to be managed. Main message hub services were fully managed in the cloud to control data transmission through queuing discipline. Database servers were deployed to keep persistent data for further usage. Overall web services to allow data access and user permissions at different user services were also managed at this layer. Finally, the data analytics were performed over the cloud to use cloud infrastructures that were limited at the edge node.

## 5. Discussion of the Case Study—Rwanda

Rwanda is an eastern African country with a fast-growing economy. Its cities have been gradually expanding, with 881,445 city dwellers in its largest cities and 8,169,341 in rural areas [40]. Increasing urban populations have also led to increasing risk of fire. There is a high risk of fire outbreak due to new public construction, combined with a lack of firefighting equipment and sufficient people to operate it. Over the last 10 years, fire outbreaks have increased, with 327 fires registered across the country since 2010. Public buildings include shops and higher learning institutions, holding a huge number of people. The case study taken used for this research was from the University of Rwanda, College of Science and Technology, Kalisimbi Block, where the research was conducted and deployed. Table 3 summarizes fire incidents recorded from 2010 to 2019.

A national inquiry, set up to investigate the causes of the fire outbreaks, found that 61% of fire incidents were due to short circuits, while 22% were due to unknown causes [41]. Additionally, 9% were arson, 4% were caused by road incidents, 3% were caused by hazardous domestic activities, and 1% were caused by chemical explosion as shown in Figure 3.

Due to the aforementioned fire outbreaks, there was a need to address the issue by incorporating an IoT solution. Some research proposed the implementation of IoT techniques for early warning and notification, but the solution was limited by system escalation and modular reusability [42]. Many applications of IoT focus on the smart health care domain [43]; however, this research proposed IoT frameworks applied in smart buildings with the following capabilities: reusability, scalability, accessibility, interoperability, and updateability.

Referring to the sensor node deployment shown in Figure 4, temperature, humidity, and CO_2_ sensors were connected in the middle of the room to maximize capture. A proximity sensor was deployed at the entry point of the room. The reason for choosing this configuration was to be sure that any captured positive value would show the moving being. The last parameter of interest was the counting of entries using two communicating infrared sensors (IR). The presence of a moving object in the office would not necessarily mean that it was a human being. All of those sensors in combination would help to improve the questions asked in [44], for instance, to recognize the presence of humans if there were a moving chair during an earthquake.

The management of the architecture used a distributed set of identical IoT devices deployed in different locations. Each device was configured to be part of the same group.

Every time a new device was turned on or connected, there were different steps to follow:-A New IoT device, such as a sensor or sensor node, is connected to the manager.-The connected device/sensor is identified with its unique identification.-The manager assigns the device to the device group, meaning the floor coverage of the building.-The latest device group configurations are pulled.-The manager checks for any updates from the broker (manager) map table to update the connected groups.

Any time the admin deployed a new device or changed existing device configurations, the following steps were to be followed:-Push a new container version, with its corresponding tags, into the configured Docker registry.Update the control API.Update the backend database with the new configuration.Update all connected IoT devices in their respective groups.

For occupant presence counting, Algorithm 1 was used to synchronize two communicating IR sensors (one counting incoming users, the other one decrementing ongoing users). Its deployment, shown in Figure 4, positioned it at any known entry points of the building. The reason for this was to facilitate the knowledge of the aggregated number of occupants in a specific building, in case of an incident.

The configuration and cooperation of two IR sensors were programmed using the following algorithm:
**Algorithm 1: An Algorithm for Entry Counting**
***Input****: peopleIN, peopleOUT, IRinStatus, IRoutStatus, inPeople, outPeople, stayPeople*
***Output****: Number of stayPeople in a Room*1***Initialization of variables****: assign **zero to** variable **peopleIN**, **peopleOUT***2***while (true) do***3 ***Read** IRinStatus*4 ***Read** IRoutStatus*5 ***if IRinStatus == 0 then***6  ***RETURN** inPeople ← peopleIN++*7 ***end if***8 ***if IRoutStatus == 0 then***9  ***RETURN** outPeople ← peopleOUT++*10 ***end if***11 *stayPeople = inPeople − outPeople*12 ***if stayPeople ≤ 0 then***13  *No people inside // No sending process as there is null data to be sent*14 ***else***15  ***RETURN stayPeople***16  *Send stayPeople in the database // Send data to the database*17 ***end if***18***end while***

### 5.1. Deployment Architecture of the IoT Sensor Nodes

Figure 5 depicts the deployment design architecture of the proposed building monitoring IoT system. The system components concerning the Edge/Gateway/Fog/Cloud architecture are also shown.

The system consisted of multiple nodes deployed in different rooms or locations to monitor selected environmental parameters, including temperature, humidity, CO_2_, motion, and occupants. The end nodes were equipped with various sensors (such as humidity, temperature, CO_2_, motion, and RIF sensors) and connected to the manager (centralized controller) to ensure connectivity with deactivation in case of poor function.

The controller was configured as a Linux computer with all management capabilities. An orchestrator was installed there to manage all microservices, including the services to connect, update, and disconnect any newly connected sensor node. The evaluation of running, pending swarm of services, was monitored through the configured services Docker Swarm visualizer. The end nodes sent the data to the cloud database using publish/subscribe services. However, the management of sensor nodes’ gateways was managed by the manager through REST services. The data analytics were done in the cloud to aggregate the data and make predictions.

### 5.2. The Representation of the IoT Domain Model

This section details the architecture of the proposed connected modules from the device end to the human services visualization.

Figure 6 represents the identification of the proposed IoT scenario. The overall figure shows that the user needed to interact with a physical entity in the physical world. In order to allow/control the interaction of humans with the rest of the system, there was an entity called an active artifact that was running software. In the context of this research, an active digital artifact role was played by the Docker manager to harmonize the sensor nodes’ operation with their respective containers. A physical entity represented a discrete identifiable entity in the physical environment: building, rooms, as well as any actuation. A virtual entity represented the physical entity in the digital world; in other words, it could be represented as the logical interface to allow physical object interaction. The device provided a medium for interaction between physical entities and virtual entities. It could be attached to the physical entities or placed closer to the physical entities. Resources were software components, either on-device or network resources. Services provided an interface for integrating with the physical entity. It accessed the resources hosted on the device (or the network resources) to obtain information about the physical entity or perform the actuation process upon it.

Figure 7 shows the process of mounting a new sensor node to the existing system. The configuration of a new sensor node consisted of only a few steps. The first step was the process of mounting the new device with its respective driver. The second was to allow the preparation of device updates on the list of available services. The third was to be able to pull the update from the container manager. The last step was to be able to interact by pushing data to the respective entities.

Due to the publish and subscribe services offered by the MQTT protocol for real-time information sharing between connected devices with the central controllers, the sensor node was able to connect with predefined rules to meet the required configuration. The broker deployed in fog edge with Docker Swarm was able to work in cooperation to ensure that there was no interference. The Docker Swarm could also make sure that every device with its resources was containerized. Every container was assigned its unique identification to allow the restriction to available resources.

Figure 8 shows the system views from the end-user perspective. The user interacted with the system by accessing its services through the web application. In the monitoring process, the end-users, such as the fire brigade, called Emergency Response Controller (ERC) in this context, must be authenticated by the system.

For the client application (services) to be able to access the data generated by sensors, the services must be subscribed to the broker to allow smooth message queueing and notification sharing using the MQTT protocol. The broker used in the research implementation was EMQ. In the data management process, all generated data from the sensor nodes were saved to another service database subscriber known as the PostgreSQL server. Data could be retrieved from the database server in a different format based on the user context. In this situation, as there was a need for data for research purposes, there was no beautified user interface built. The data were kept for further use and retrieved in JSON format for preprocessing. 

For performance monitoring, a combination of different monitoring tools were used—cAdvisor, Prometheus and Grafana. Figure 8 shows only Prometheus to simplify the view. All services used in this research (subscription/publish, web application, web server, database server, monitoring servers) were built as microservices under the Docker container. This configuration eased dependencies on management and ensured the correct configuration. The services, known as microservices, were wrapped into an efficient orchestration module known as Docker Swarm. This facilitated the process of activation and deactivation any time there were any malfunctioning services, including device services.

Configuration of the services used in this research was done in layering structures. Prometheus was configured to ensure all the metrics from the cAdvisor were ingested and grabbed. To confirm the time interval of sharing the data, the scrape interval was set, including the endpoint where the cAdvisor was executing. It was forced to expose the data at port 8080.

All defined services were wrapped within Docker containers to embed all required dependencies.

The configurations in the Docker-compose file specified which containers were part of the installation and which ports were exposed by each container service. The configuration file focused on data ingestion from cAdvisor, Prometheus, and Grafana for simplicity.

One key point to mention regarding the configuration: by configuring the specific container image, the dependency was able to ensure the direction the data took from container service one to container service two, and so on. There was also another configured time-series database (TSDB) to capture metrics parameters in real-time. In this research context, the configured storage was known as the Redis database.

## 6. Results

This section describes the results and outcomes of the implementation. It details the deployment place, along with the setup of the testbed, including the service usage visualization.

### 6.1. Deployment Location Selection

This research detailed its application in the fire brigade domain. The architecture was deployed in the KALISIMBI building of the University of Rwanda (UR), as shown in Figure 4. The site was selected due to multiple fire outbreak cases noted in the student kitchen in 2018. The KALISIMBI Block site was selected because it contained many laboratory equipment and supplies for chemistry, physics, and electronics, as well as computer labs. At any time, these chemical components could have reacted and impacted students’ lives. The deployment for testing was configured on the third floor in one room, to help prevent and forecast incidents.

### 6.2. Hardware Setup

The real implementation of the proposed IoT architecture system used the hardware components detailed in Table 1 and Table 2 described above. The testbed was configured with all selected sensors based on the case study. They were all mounted in an RPi microcontroller board running Docker services. The reason for deploying all sensor nodes services as microservices in the Docker environment was to allow easy scalability with lightweight performance management capability.

The system was powered by a power source and connected to the LAN network of the campus to ensure that it could sync services with Docker Hub. All data were sent by subscribed sensor nodes through the pipelined microservices up to the hosted database in the UR campus data center. Each RPi sensor was categorized into groups to manage its resources and connection capabilities. In the testbed, the computer was used where the Docker manager was configured to facilitate microservices management and scalability, referred to in Figure 9 and Figure 10.

### 6.3. User Application Section

The user was able to access the platform services through the developed web application under JavaScript, HTML5, Cascading Style Sheets (CSS), and ReactJS technologies. More clearly, the ReactJS framework was used to develop the web application following the responsive paradigm. For monitoring of the statuses of the containers through the container manager, Docker Swarm visualization tools were used, as shown in Figure 11.

In a test experiment, the initial condition of the visualizer service was executed by the computer system. The Docker Swarm was configured in the board first. However, due to the limited capability of the storage, the laptop was used as a Docker Swarm manager. The RPi board was configured as the worker. It was given the IP address to be able to run independently.

In summary, the system consisted of one Swarm manager (laptop), and one node worker for testing purposes. All the container services were running in the Swarm manager, while the sensor node services were running on the worker side. The Swarm manager ran all services including worker services. The reason for this configuration was so that it would be able to update the new connected worker and show the status of non-performing services, as shown in Figure 11.

Figure 12 shows the behavior of the Docker Swarm manager during the process of orchestration at the scale-up process.

The other testing process was to measure the system availability level based on scalability capability for the system by trying to scale up and scale down all the deployed container services running in the Docker Swarm manager. The performed operation was designed to test the capability of the system availability when it came to serving an instant number of incoming requests that may affect the system overload.

The scaling test was designed to observe how the system would behave in cases where there were hundreds of thousands of connected workers (sensor nodes) interacting with the services. The other concern was to see how the platform behaved when many end-users requested the services (building control units).

## 7. Performance Analysis: Reliability and Scalability

This section covers the performance evaluation of the proposed framework by following the approach and metrics discussed in Section 3.2.4. The evaluation strategy used consisted of a function to generate a huge amount of data ingesting into the system to observe its behavior. The data generator function’s purpose was to simulate IoT devices as publishers. The ingested data allowed observation of how the containers would be orchestrated under the governance of the Docker Swarm manager to compete for te available resources. The goal of this experiment was to assess the proposed IoT framework from two different points of view:-**Reliability**: the reliability of the system was measured by utilizing all the available resources to ensure that the system was running at full capacity. Those resources included CPU and active memory usage, with how many data were processed in a specific period. CPU usage metrics indicated average OS core utilization for all available CPU physical and virtual cores of the running system. Active memory showed the number of bytes stored in memory during the experiment at the time of the test.-**Scalability**: the measure of the prediction for extending the system to support hundreds of thousands of IoT devices publishing data with minimum system overhead to help time-critical applications. The measurement of these characteristics was indicated by the system throughput and response time metrics, showing how the system scalability was maintained.

The excellent performance was shown by the high throughput rate, as in principle, it should have been increased linearly as the number of devices or requests increased. The calculation of the throughput was given by the number of messages received and processed by the IoT system in one second; the lower the response time, the faster the response and processing of the system.

### 7.1. Evaluation Approach and Tools

The experiment to access the performance of the proposed framework was separated into two categories. The first was from the normal connected IoT devices’ viewpoint. The second was from the simulated requests to capture the system behaviors. For the normally connected sensor node, the measurements were also acquired. Testing was done to ensure the experimental testbed was feasible. The simulated experiment was done to test the performance while system scaling was conducted.

During the testing phase, the function of sending requests was executed to simulate virtual IoT devices published in the specified time interval. The simulated IoT devices sent requests or messages with an increment of 10^3^ (1000, 100,000, 1,000,000) to the IoT platform. In this experiment, the platform ran under the MQTT publishing operation with Quality of Service (QoS) level 0—the client simply published the message, and there was no acknowledgment by the broker. The reason for choosing level 0 was to avoid the system overhead created by acknowledgment messages. The data used for the system ingestion were simulated and represented an observation of a unique and random phenomenon. The estimated average size of each datum generated was 240 bytes. These implied the amount of data ingested to the platform backend services from sensor nodes and container controller.

The experiment aims to collect CPU, throughputs, and memory metrics usage is conducted by using cAdvisor as container resources usage analysis platform and Prometheus in combination with Grafana for monitoring the proposed platform as shown in Table 4.

### 7.2. Experiment Results

This subsection details the performance analysis conducted given different parameters. The key concerns were related to the reliability and scalability of the system based on the changing requests. 

#### 7.2.1. Performance in Terms of System Reliability and Scalability

Figure 13 and Figure 14 show different metrics captured for performance evaluation in case the real IoT sensor node device were to be sending data. In the section on CPU usage, the average performance results showed that the maximum usage was 2.25 while the minimum usage was 0.25. This was considered a benchmark against which to measure if any increase in the load could affect CPU usage.

The evaluation showed that, due to the use of microservices, there was no single core overload. The graph showed that all works were distributed among the available 4 cores used in this testing, with an average of around 40% (0.4). The engagement of all cores implied an increase in system throughput.

On the graph of throughput, the dominant operation was the reception as the data were being sent from the sensor node to the platform. The observed maximum value of the reception was 46,000 bytes per second. It showed a minimum of around zero at some specific period. The presence of some points with zero throughputs could depend on the sensor node device set to send data at an interval of 5 s. The maximum memory usage in this configuration was around 4600 megabytes, while the minimum was around 4200 megabytes. This implied that the average usage of 4400 megabytes of memory.

Figure 15 and Figure 16 depict the observations of the system behavior after applying the generated IoT sensor requests. In predicting the platform scalability, 10,000 IoT sensor requests were generated. In these graphs, there was a considerable increase in CPU usage with an average of (0.75) 75%. The memory usage maximum was around 4800 megabytes while the minimum used was around 4400 megabytes. This implied an average of 4600 megabytes of memory. In this experiment, the throughput graph showed a maximum of around 50,000 bytes/second with a minimum of around zero, generating an average of 25,000 bytes/second.

Figure 17 and Figure 18 show the considerable increases, especially in CPU core usage, with the generation of 1,000,000 requests. The average usage of all CPU cores was around (0.8) 80%. The maximum memory of 5000 megabytes and the minimum of 4500 megabytes of RAM were consumed. The average memory usage as around 4750 megabytes. The maximum throughput measured was equal to 60,000 bytes per second, with an average of 30,000 bytes per second.

#### 7.2.2. Performance Measurement in Terms of Container’s Resources Usages at the Scale-Up

The second experiment was conducted by increasing the number of nodes to track how the system behaved if it were scaled up. This experiment was conducted by setting the number of nodes to 3; the observations were measured using the same performance tools discussed (Prometheus and Grafana). In this experimental setup, the assumption was to scale up the containers to observe the system behaviors. The experiment was detailed with two entries: scaling the application, in terms of container services, and generating different requests to observe the system. In this experiment, the observations are sticking to CPU, Memory, and Traffic metrics behaviors.

Figure 19 visualizes the memory and CPU usages in the scaled 3 nodes, using 1000 generated requests. The graph of memory usage showed that the increases in requests could increase the amount of consumed memory, but for the time being, there was no increase. The CPU usage graph showed that, while at the time, there was a small increase in usage, this increase may have depended on the internal cooperation of all container services involved in the data processing.

The performance metrics were concerned with the transmission cost of the network as shown in Figure 20. The main point of tracking those parameters was to observe how the internal containers were costly while exchanging message requests. In network transmission, the busy container was cAdvisor, as it was concerned with the huge number of requests being generated.

Figure 21 shows the visualization of the system after increasing the number of requests up to 1,000,000. The graph shows that CPU and memory did not change significantly, but the trend increased compared to Figure 19.

Figure 22 shows the increase in performance for the network transmission, as this experiment did not discuss testing the user access of the requests at the user end side.

Figure 21 and Figure 22 demonstrated the overall behavior of the platform using 3 nodes. Figure 21 shows the increase of the CPU usage as the number of requests increased.

Figure 23 shows the CPU usage performance, while Figure 24 visualizes the performance of the system in terms of memory usage and network transmission. The graph summarized the details shown in Figure 19, Figure 20, Figure 21 and Figure 22. It was observed that the memory did not increase in consideration of the increase of the requests, as seen in the observations detailed in the above figures. The network transmission at the receiver side (Rx) did not increase much as the experiment performed was aimed at generating requests to be sent to the platform. Generating requests trying to access the generated data was out of the scope of this research.

The network transmission at the transfer side showed tremendous changes, but it was also observed that, at one hundred requests, the graph began to go down to ten thousand, but then increased.

## 8. Conclusions

The implementation of a comprehensive solution for the whole IoT lifecycle was presented in this research. The lifecycle presented fit into four phases: data capture, communication, analysis, and implementation. A generic framework architecture, based on the above layers, was defined in the DevOps paradigm. The architecture complied with the main system requirements of all proposed IoT solutions.

The proposed IoT solutions were discussed in the building fire prevention sector, to improve building comfort and security. Different challenges were discussed concerning the scalability, updating, and performance of IoT devices with low computing capabilities. The IoT devices were installed in buildings, accessing the building’s electrical power resources.

In the proposed framework for data transmission and management, the nonfunctional requirement (QoS) was considered to address all critical cases to be solved by IoT solutions. It also showed that some computation was possible at the edge side, minimizing the decision time delay.

Data transmissions were considered by ensuring data transfer in real-time using MQTT-based protocols, along with data packaged in JSON format—which has been considered lightweight—to ensure low network bandwidth consumption. For services, scalability, and reusability, the container-based and microservices solution was proposed, using Docker. It satisfied the horizontal scalability of the framework. The sensor nodes, considered Docker workers for updating and easy connectivity, were under the control of the Docker Swarm manager. Data sent from sensor nodes were managed by the Nginx web server, which also acted as a load balancer in case of the huge amount of user requests coming from the front-end.

The IoT framework presented in this research was validated and used in a smart building scenario to contribute to fire prevention situations. The system was deployed in the UR–KALISIMBI block site for three months to experiment with the challenges of the data capture process. The performance metrics were extracted to visualize if they were suitable for adoption by the whole University.

Finally, for future work, the authors will improve the IoT interoperability for disparate and heterogeneous sensor devices, data, and the standard IoT models. Additionally, the extensions will concentrate on an advanced custom visualization layer with machine learning techniques included for decision making.

## Figures and Tables

**Figure 1 sensors-22-06462-f001:**
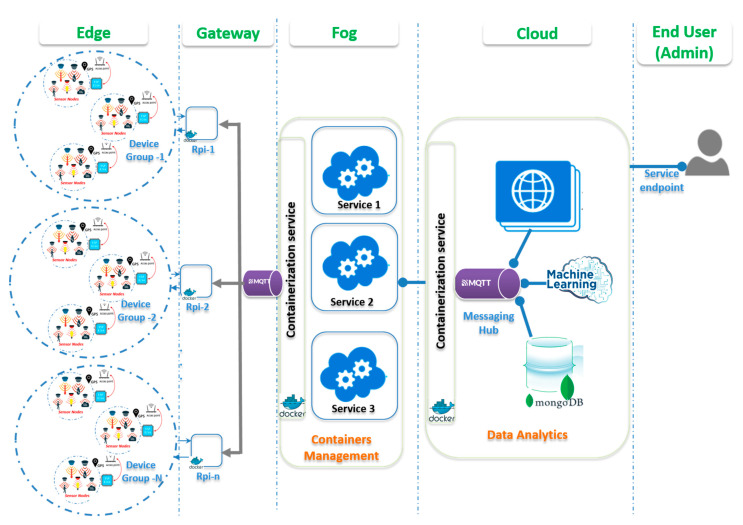
Proposed Deployment Architecture.

**Figure 2 sensors-22-06462-f002:**
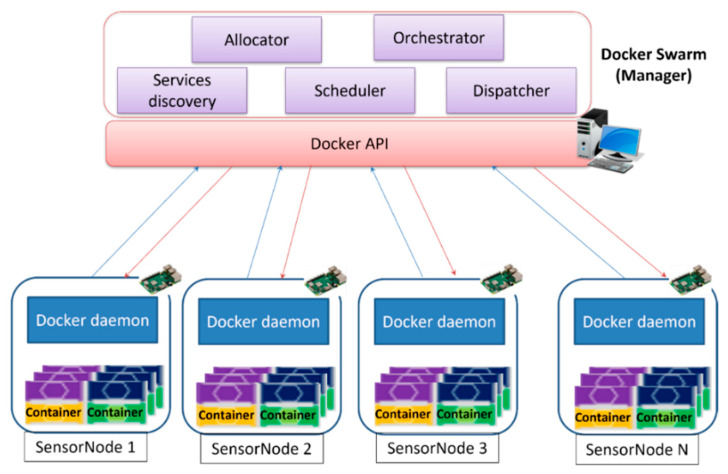
Architecture with containerization principles.

**Figure 3 sensors-22-06462-f003:**
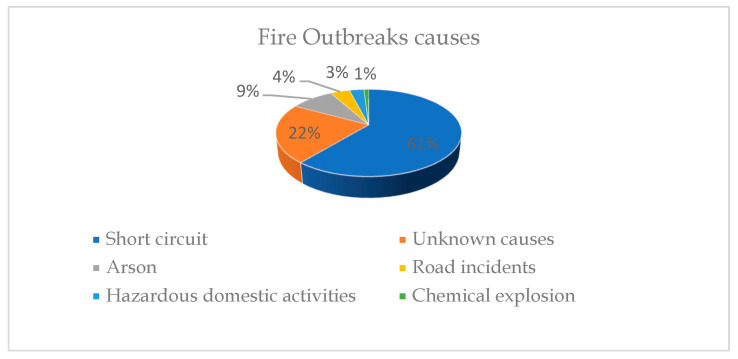
Fire outbreak causes.

**Figure 4 sensors-22-06462-f004:**
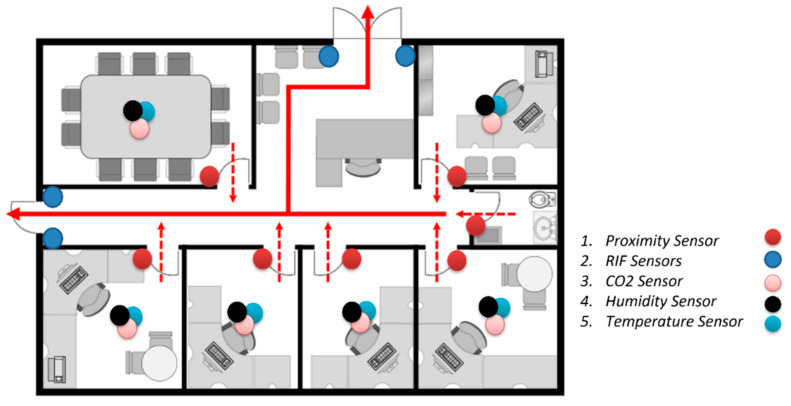
Sensor Deployment architecture in UR, Kalisimbi Block offices.

**Figure 5 sensors-22-06462-f005:**
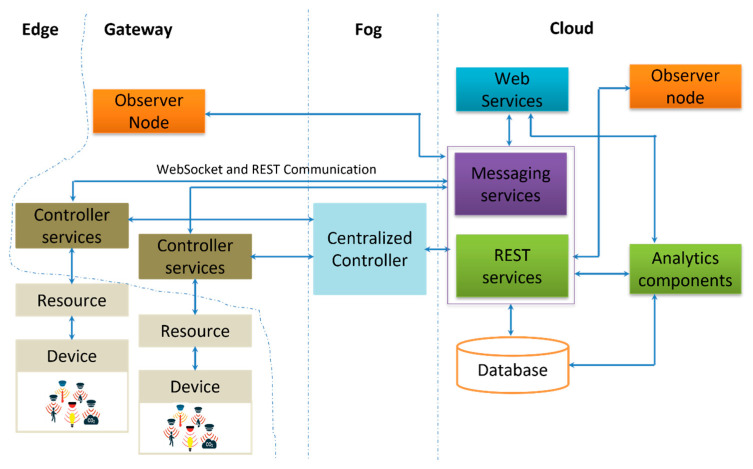
Deployment design of building monitoring IoT system.

**Figure 6 sensors-22-06462-f006:**
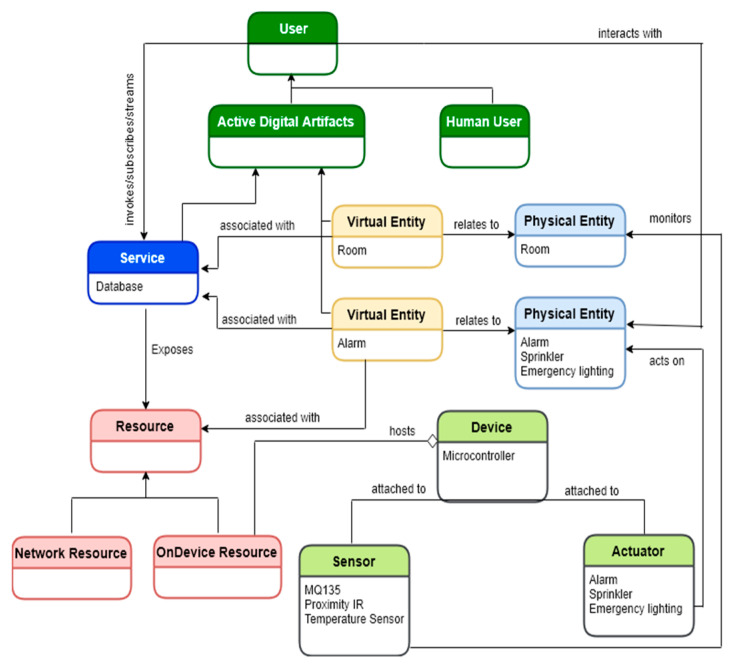
Proposed UML representation of the IoT Domain Model.

**Figure 7 sensors-22-06462-f007:**
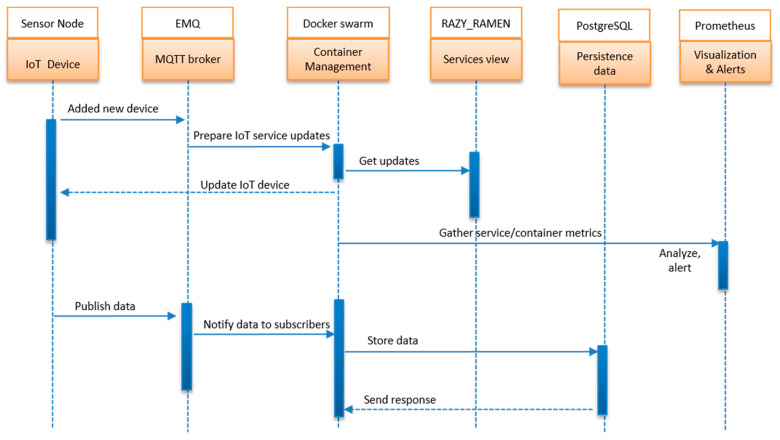
Sequence diagram activity at the sensor node perspective.

**Figure 8 sensors-22-06462-f008:**
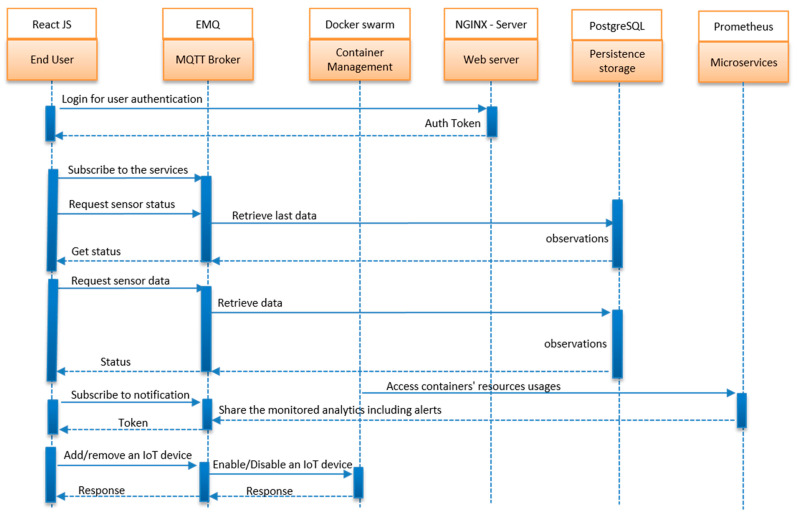
Sequence Diagram for End-User Perspective.

**Figure 9 sensors-22-06462-f009:**
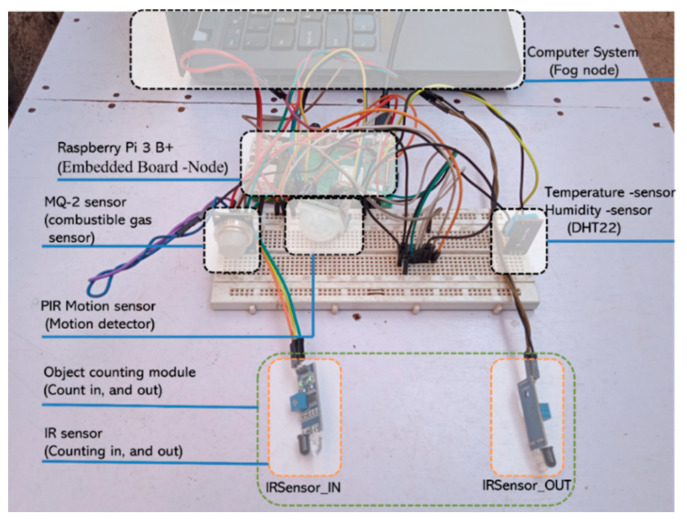
Detailed mounted sensor node with all related components.

**Figure 10 sensors-22-06462-f010:**
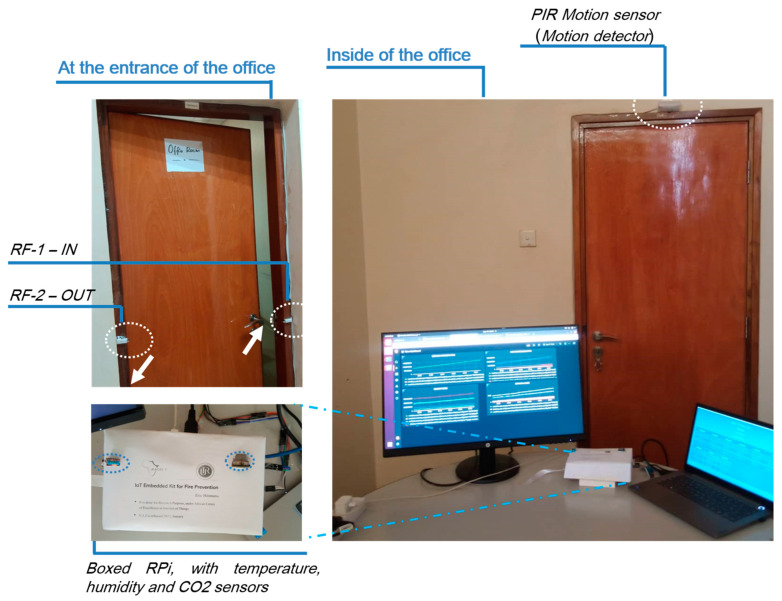
Real mounted sensor nodes with all related components in the building.

**Figure 11 sensors-22-06462-f011:**
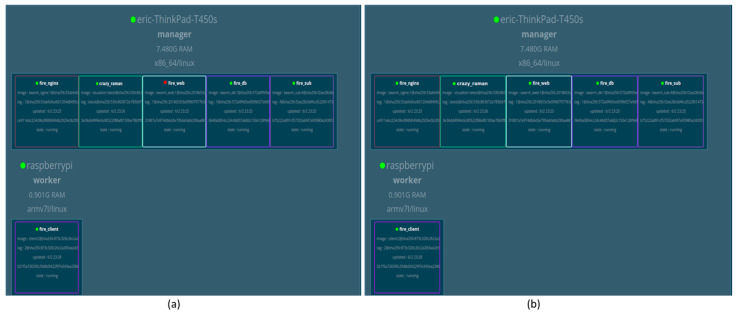
(**a**) Visualization of container clusters with one non-working container service with the attached RPi worker. (**b**) Visualization after system container stabilization.

**Figure 12 sensors-22-06462-f012:**
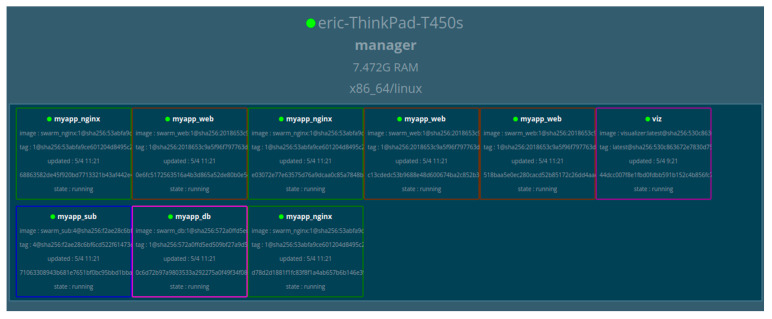
Docker Swarm with duplicated container services at the scale up purpose.

**Figure 13 sensors-22-06462-f013:**
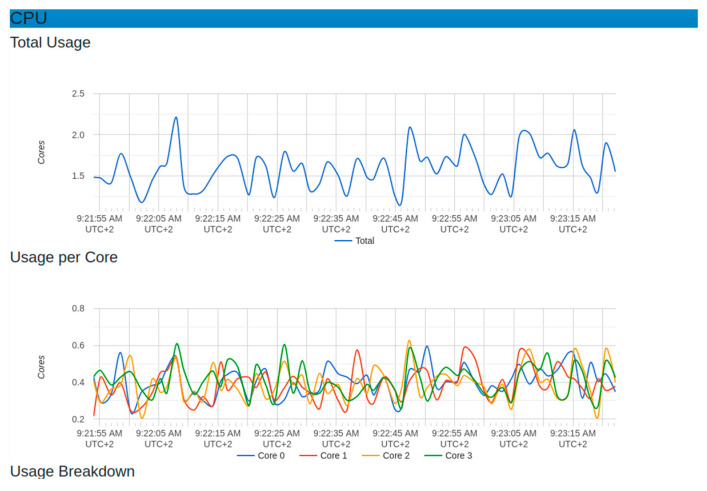
CPU resource behavior with normal sensor node setup.

**Figure 14 sensors-22-06462-f014:**
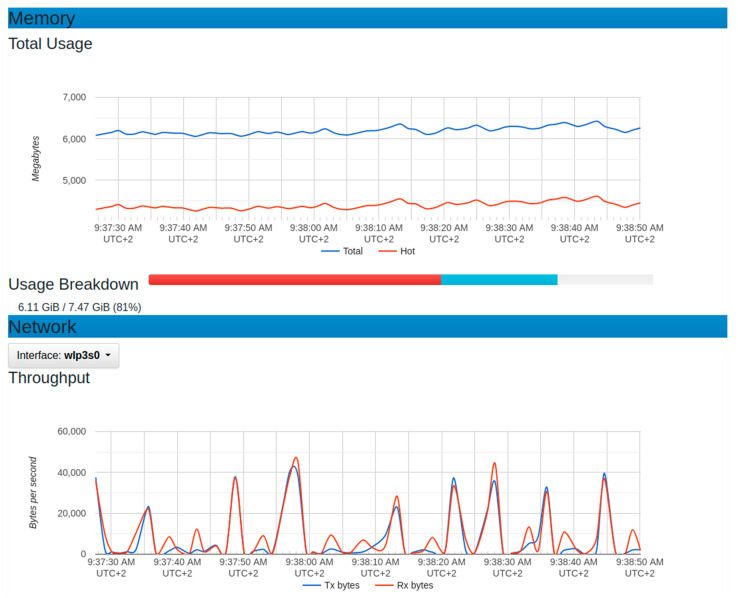
Resource usage for memory and throughput with normal sensor node setup.

**Figure 15 sensors-22-06462-f015:**
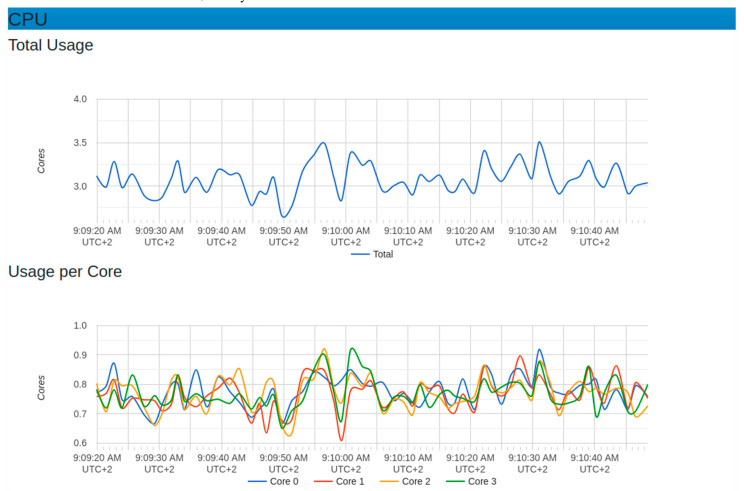
CPU resources with 10,000 generated sensor requests.

**Figure 16 sensors-22-06462-f016:**
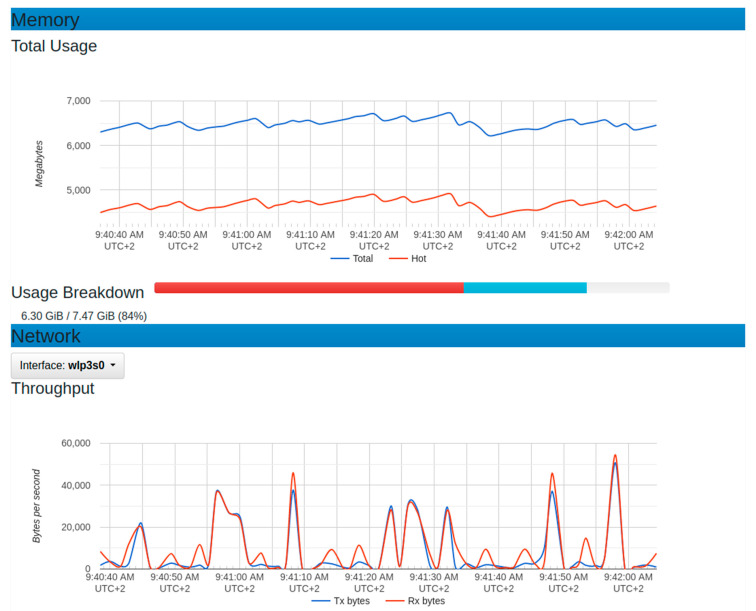
Resource usage for memory and throughput with 10,000 generated sensor requests.

**Figure 17 sensors-22-06462-f017:**
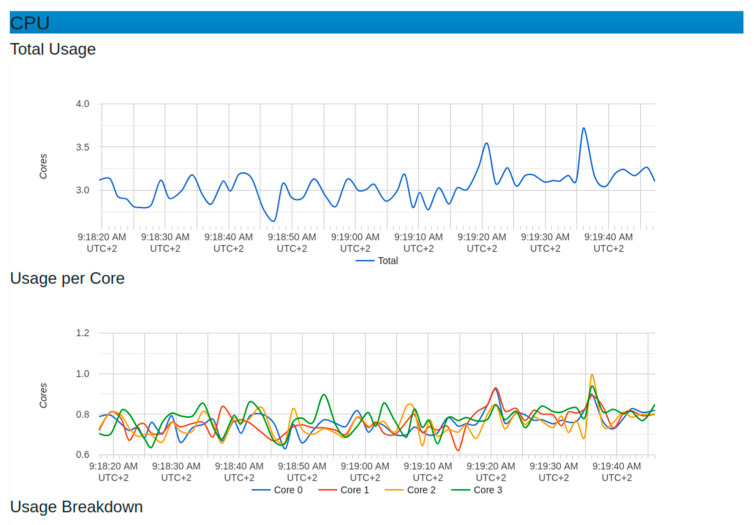
CPU resource with 1,000,000 generated sensor requests.

**Figure 18 sensors-22-06462-f018:**
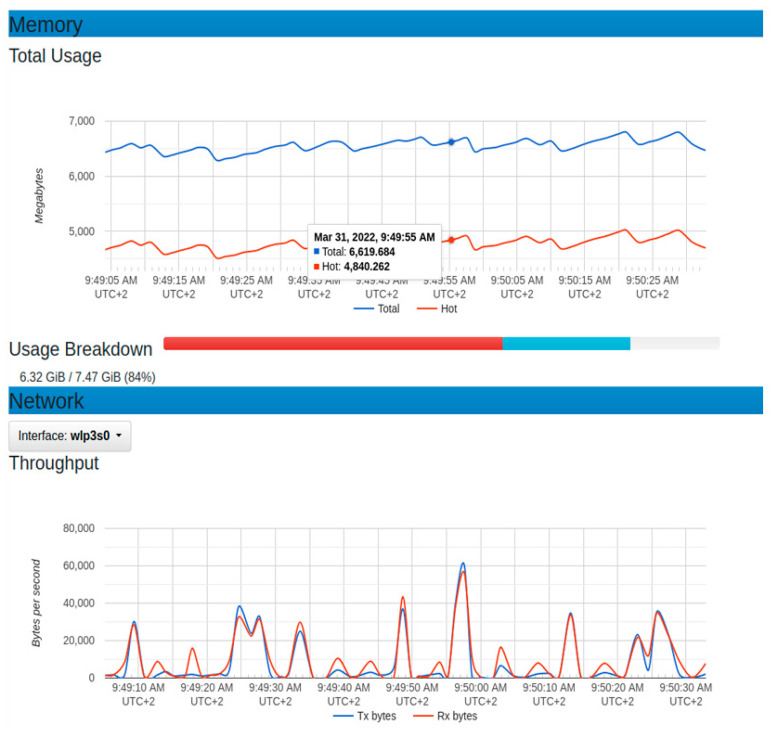
Resource usage for memory and throughput, with 1,000,000 generated sensor requests.

**Figure 19 sensors-22-06462-f019:**
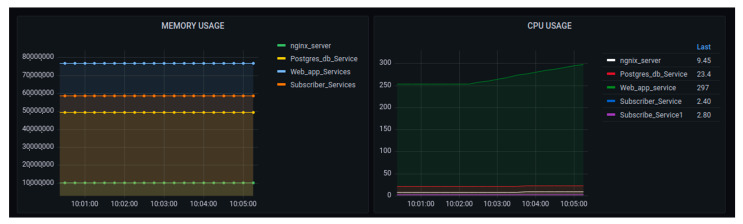
Performance of the platform after scaling up with the requests of 1000—memory and CPU usage.

**Figure 20 sensors-22-06462-f020:**
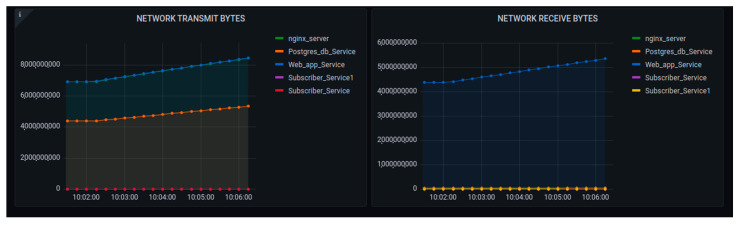
Performance of the platform after scaling up with requests to 1000 network data transmissions.

**Figure 21 sensors-22-06462-f021:**
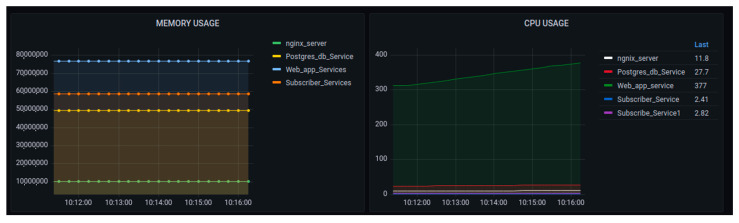
Performance of the platform after scaling up with 1,000,000 requests: memory and CPU usage.

**Figure 22 sensors-22-06462-f022:**
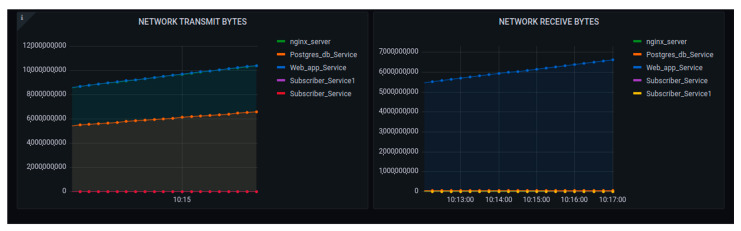
Performance of the platform after scaling up the requests to 1,000,000 network data transmissions.

**Figure 23 sensors-22-06462-f023:**
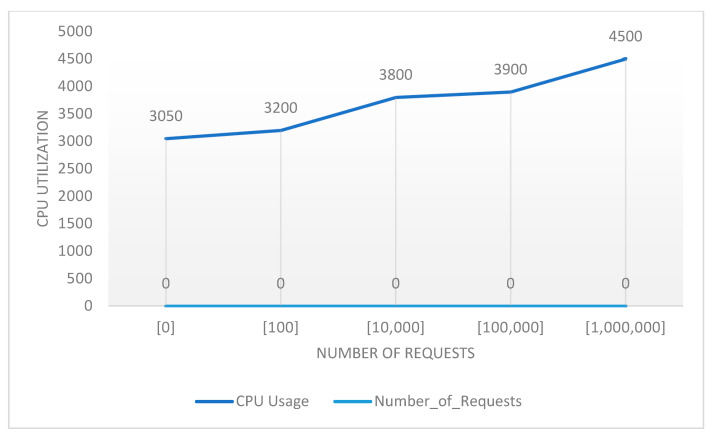
Performance of the scaled system by 3 nodes: CPU usage.

**Figure 24 sensors-22-06462-f024:**
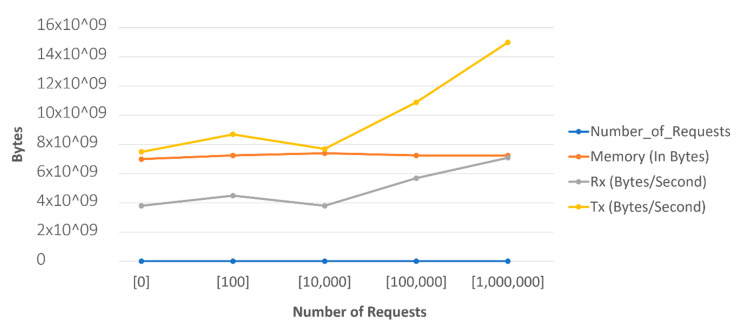
Performance of the scaled system by 3 nodes: memory and transmissions (Rx, and Tx).

**Table 1 sensors-22-06462-t001:** Configuration of the experimental testbed.

Device	Desktop	Raspberry Pi 3 B+
CPU	4.5 GHz Intel Core TM (4 cores)	Quad core Cortex-A72 (ARM v8), 1.5 GHz
Memory	4 GB DDR3	1 GB LODDR4 RAM
Storage	500 GB HDD	32 GB (microSD card)
OS	Linux x86_64	Raspbian GNU/Linux (Jessie)
Installed Software	Docker Swarm, NGINX, PostgreSQL, Redis, cAdivisor, Prometheus, Grafana	Docker, python

**Table 2 sensors-22-06462-t002:** Sensor’s operation ranges.

Name	Description	Min Value	Max_Value	Min_Distance	Max Distance
DHT11	Temperature	0 °C ± 2 °C	50 °C ± 2 °C	1 m	10 m
	Humidity	20% ± 5%	80% ± 5%		
PIR	Motion sensor	0 V [0]	3 V [1]	1 m	7 m
IR	Obstacle sensor	0 V [0]	5 V [1]	1 m	5 m
MQ-2	CO_2_ and Smoke sensor	300 ppm	10,000 ppm	2 cm	4 m

**Table 3 sensors-22-06462-t003:** Summary of the fire incidents recorded over the past 10 years.

Years	Alerts	Deaths	Injured	Houses Destroyed	Crops Ha	Liver-Stock	Churches	Admin Buildings
2010	0	4	9	73	5	0	0	0
2011	0	2	7	84	12	0	2	0
2012	0	0	8	83	21	1	3	0
2013	0	8	8	88	17	2	3	1
2014	0	3	7	88	12	0	6	0
2015	0	0	0	0	0	0	0	0
2016	45	2	7	37	40	0	0	0
2017	48	2	12	33	26	3	5	0
2018	0	0	0	15	4	0	2	1
2019	0	1	19	30	4	0	11	0
Total	93	22	77	531	141	6	32	2

**Table 4 sensors-22-06462-t004:** Specifications of environments used for performance analysis.

Specifications	Test Runner	Deployment Server
CPU	4.5 GHz Intel Core TM (4 cores)	4.5 GHz Intel Core TM (4 cores)
Memory	8 GB DDR3	8 GB DDR3
Storage	500 GB HDD	500 GB HHD
OS	Linux x86_64	Ubuntu 20.04 LTS
Installed Softwares	Docker Swarm, MQTT Microservices, NGINX, PostgreSQL 10.3, Redis, cAdivisor, Prometheus, Grafana	Docker Swarm, MQTT, Microservices, NGINX, PostgreSQL 10.3, Redis, cAdivisor, Prometheus, Grafana

## Data Availability

All data used to achieve the research objectives are available online (ACEIoT portal at the University of Rwanda).

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
