# Peer review of "Containerized Architecture Performance Analysis for IoT Framework Based on Enhanced Fire Prevention Case Study: Rwanda"

_sensors, 2022, doi:10.3390/s22176462_

Round 1

Reviewer 1 Report

The paper is well-written apart from minor language problems that need to be fixed e.g.,

-          “the gap between product design with its operation…” (with->and),

-          Line 77-78, too many citations in one sentence, also in random places

-          Line 160, “…executed” -> are executed

-          Line 229, remove full stop from titles

-          Line 238, “other than” -> instead of

Other presentation problems concern the readability of figures 12 to 17: even in 150% zoom of the pages, I could not get the individual legends (colors and values) of each graph. If this is not important, why included?

More important, in the abstract you mention, beyond safety, the problems of energy-saving, and comfort upgrades, but those two are not discussed or presented/demonstrated in the paper. This is a misleading reference. Please remove any references to ‘comfort’ or ‘energy-saving’ in the article.

The Introduction section is too long with no clear structure: too many paragraphs introducing technologies but there are no explicit statements regarding the aim, objectives, motivation and contribution of the presented work. Also, what are the limitations of related work on the specific problem, and how is your proposed framework overcoming them? These I would expect to read in this section, and I would suggest to revise it towards this approach.

In general, although it is good to provide implementation details of proposed solutions, it seems that this paper has overdone it, in contrast to a more critical discussion on the innovation and purpose of the proposed framework. So, the feeling I get is that, although the approach uses latest technologies to improve performance and scalability of IoT solutions, there is a lack of research questions answered, that other approaches cannot! Also, what is missing (left for future work I guess) is an advanced visualization layer, probably related to a visual analytics component also, and on the other hand, an interoperability layered approach (e.g., using standard or semantic IoT models for representing disparate and heterogeneous sensor data).

Based on these findings, I would suggest a revision, emphasizing on the innovation part of the approach, explicitly stating the contribution of this work beyond the state-of-the-art.

Reviewer 2 Report

1. In this paper, a containerized architecture  framework was proposed to support multiple concurrent deployed IoT applications in the smart building. In addition, a complete study was demonstrated.

2. This paper is well-organized. Also, I think it is interesting to readers investigating IoT application.

3. The revolution of algorithm 1 should be improved.

4. The indexes of references should be referred according to the orders of appearance in this paper.

Reviewer 3 Report

The manuscript is mainly about the design, development and application for the IoT-based intelligent building, especially with the fire prevention application; it comprehensively explains the relevant technologies, hardware devices, platforms, architectures, etc., and is helpful to the researchers and developers of the various kinds of IoT-based applications.

The organization of the manuscript is reasonable, and the English expression is smooth. It is recommended to publish after minor revision

Several comments or suggestions:

1, The manuscript is mainly about the integrated development and application of many existing technologies, the research of new theories and/or methods is relatively lacking;

2, On Line 485, Table 1 should be Table 3, and on Line 766, Table 2 should be Table 4;

3, In Figure 9, it is better to present a real building application case of the various kinds of sensors.
